# Comparative Cyto-Histological Genetic Profile in a Series of Differentiated Thyroid Carcinomas

**DOI:** 10.3390/diagnostics14030278

**Published:** 2024-01-27

**Authors:** Maria de Lurdes Matos, Mafalda Pinto, Marta Alves, Sule Canberk, Ana Gonçalves, Maria João Bugalho, Ana Luísa Papoila, Paula Soares

**Affiliations:** 1Department of Endocrinology, Diabetes and Metabolismo, Centro Hospitalar Universitário de Lisboa Central, Hospital Curry Cabral, 1050-099 Lisbon, Portugal; 2Instituto de Patologia e Imunologia Molecular da Universidade do Porto (IPATIMUP), i3S—Institute for Research & Innovation in Health, 4200-135 Porto, Portugal; mafaldap@ipatimup.pt (M.P.); scanberk@ipatimup.pt (S.C.); 3Gabinete de Estatística do Centro de Investigação do Centro Hospitalar Universitário de Lisboa Central, EPE, Nova Medical School, 1169-045 Lisbon, Portugal; marta.l.alves@gmail.com (M.A.); ana.papoila@nms.unl.pt (A.L.P.); 4Centro de Estatística e Aplicações da Universidade de Lisboa (CEAUL), 1749-016 Lisbon, Portugal; 5Department of Pathology, Centro Hospitalar Universitário de São João, 4200-319 Porto, Portugal; u018610@chsj.min-saude.pt; 6Department of Endocrinology, Centro Hospitalar Universitário de Lisboa Norte, Hospital de Santa Maria, 1649-028 Lisboa, Portugal; maria.bugalho@chln.min-saude.pt; 7Medical Faculty, University of Lisbon, 1649-028 Lisboa, Portugal; 8Medical Faculty, University of Porto, 4200-135 Porto, Portugal

**Keywords:** ultrasound-guided fine needle aspiration cytology (US-FNAC), *TERT*, *BRAF*, *RAS*, genetics, differentiated thyroid carcinomas (DTCs), indeterminate nodules, papillary thyroid carcinomas (PTCs)

## Abstract

Introduction: Molecular tests can contribute to improve the preoperative diagnosis of thyroid nodules. Tests available are expensive and not adapted to different populations. Aim: This study aimed to compare the cyto-histological genetic profile and to evaluate the reliability of molecular tests using ultrasound-guided fine needle aspiration cytology (US-FNAC) in accurately diagnosing differentiated thyroid carcinomas (DTCs) and predicting biologic behavior of papillary thyroid carcinomas (PTCs). Materials and Methods: The series included 259 patients with paired cyto-histological samples totaling 518 samples. The genetic alterations were analyzed via PCR/Sanger sequencing. The association with clinicopathologic features was evaluated in PTCs. Results/Discussion: From the 259 patients included, histologies were 50 (19.3%) benign controls and 209 (80.7%) DTC cases, from which 182 were PTCs; cytologies were 5.8% non-diagnostic, 18.2% benign, 39% indeterminate, and 37.1% malignant. In histology, indeterminate nodules (*n* = 101) were 22.8% benign and 77.2% malignant. Mutation frequencies in cytology and histology specimens were, respectively, *TERTp*: 3.7% vs. 7.9%; *BRAF*: 19.5% vs. 25.1%; and *RAS*: 11% vs. 17.5%. The overall cyto-histological agreement of the genetic mutations was 94.9%, with Cohen’s k = 0.67, and in indeterminate nodules agreement was 95.7%, k = 0.64. The identified mutations exhibited a discriminative ability in diagnosing DTC with a specificity of 100% for *TERTp* and *BRAF*, and of 94% for *RAS*, albeit with low sensitivity. *TERTp* and *BRAF* mutations were associated with aggressive clinicopathological features and tumor progression in PTCs (*p* < 0.001). The obtained good cyto-histological agreement suggests that molecular analysis via US-FNAC may anticipate the genetic profile and the behavior of thyroid tumors, confirming malignancy and contributing to referring patients to surgery.

## 1. Introduction

Thyroid nodular disease is very common, and distinguishing benign from malignant nodules is still a major challenge in clinical practice. Thyroid nodules are diagnosed using palpation in 5% of women and 1% of men in iodine-sufficient parts of the world [1]. Using thyroid ultrasound, nodules can be identified in 19% to 68% of the individuals [2]. Thyroid cancers are diagnosed in 1% to 5% of nodules [3], with more than 90% being differentiated thyroid carcinoma (DTC) [4].

The incidence of DTC, out of which papillary thyroid carcinoma (PTC) is the most prevalent (over 95%) [5], has increased in the last years, mainly attributed to enhanced sensitivity of complementary diagnostic techniques and environmental factors, but not accompanied by a corresponding change in mortality rate [6,7,8]. Despite its favorable prognosis, the idea has been advanced that clinicopathological features and the presence of some genetic alterations can influence their progression [9]. Accurate pre-surgical diagnosis is crucial to decide the optimal medical or surgical treatment [10].

Ultrasound-guided fine needle aspiration cytology (US-FNAC) is central in the diagnosis of nodular thyroid disease, but up to 30% of nodules remain without a definitive diagnosis [11]. Bethesda II and VI cytological results have excellent performance when assessing the benign or malignant nature of a thyroid nodule. Diagnostic limitations are particularly significant in the non-diagnostic (ND) and in indeterminate Bethesda categories III, atypia of undetermined significance (AUS), and IV, follicular neoplasm (FN) [12]. Each category harbors a different risk of malignancy and a different option for treatment and follow-up [13,14].

Genetic changes in genes encoding cell signaling pathways determine tumor development, including in DTCs. The main known genetic causes of thyroid cancer include point mutations in the *BRAF*, *RAS*, *TERT*, *RET*, and *TP53* genes and the fusions genes *RET/PTC*, *PAX8/PPAR-Y*, and *NTRK*. Numerous studies have been conducted to identify genetic or molecular biomarkers that facilitate the pre-surgical diagnosis (in categories ND, III, and IV) and prognosis of thyroid cancer, but current results remain inconclusive and without definitive evidence [15,16].

Thyroid surgery remains the only diagnostic (and usually curative) approach for suspicious and malignant nodules. However, the inability to differentiate, preoperatively, between benign and malignant nodules among those classified as Bethesda III or IV hampers the ability to prevent unnecessary surgeries and the subsequent reduction of possible complications [17].

In 2017, the European Thyroid Association (ETA) presented guidelines for molecular diagnoses using US-FNAC with thyroid nodules after reviewing methodological aspects and limitations of molecular diagnoses in thyroid cytology. Molecular tests have the potential in clinical practice for diagnosis (pre-surgical markers) and follow-up of thyroid nodules (post-surgical markers) if they are performed in specialized laboratories and with adequate calibration and analytical validation before being implemented in clinical practice [18].

Although American and European thyroid associations [19,20] consider the inclusion of molecular tests with US-FNAC, no consensus is yet achieved on the best molecular panel to use in the diagnosis of thyroid nodules. Available molecular tests have been developed in United States of America, and it is important to assess whether such tests can be used in the European population [21,22]. The most significant disadvantage of commercially available tests is related to their cost, as they are not supported by national health systems in Europe where, in many countries, their price is equal to or greater than a thyroidectomy; furthermore, to perform these tests, highly specialized reference laboratories are required [23,24].

Studies have emerged on the use of artificial intelligence (AI) in the diagnosis and classification of thyroid nodules, constituting a new area of interest in scientific research, but still with no practical applicability [25,26].

The high prevalence of thyroid nodules [27,28], coupled with the rising incidence of thyroid neoplasm, urges the exploration of strategies to prevent unnecessary surgeries while effectively treating patients. This study aims to compare the cyto-histological genetic profile by matching cytology and histology samples and to evaluate the reliability of molecular tests in preoperative diagnoses to improve the diagnostic accuracy of US-FNAC in indeterminate nodules.

## 2. Materials and Methods

### 2.1. Study Design

This retrospective study was conducted in a series of patients with thyroid nodular disease, with suspicion of malignancy, that underwent surgery, in a single non oncologic hospital between 2013 and 2020. Criteria of suspicion and/or malignancy in thyroid nodules were based in clinical ultrasound features and Bethesda classification for cytology (risk of malignancy) [11], in accordance with 2015 ATA and 2017 ETA guidelines. Recommendations for the extent of surgical treatment and surgical protocols used also followed these guidelines [18,19].

Inclusion criteria: patients diagnosed, after surgery, with differentiated thyroid carcinoma (209 cases) or benign thyroid nodules (50 controls), of both genders, and over 18 years old, with available representative histological and cytological samples.

Exclusion criteria: patients with toxic goiters or papillary carcinoma smaller than 1 cm unless metastatic, and patients with other than thyroid neoplasm at the time of thyroid surgery.

A total of 518 samples (259 histology samples and corresponding 259 US-FNAC), representing 259 patients, along with clinical information, were included in the study. Epidemiological and clinicopathological features of patients and tumors were gathered based on the information available in the clinical and histopathological reports from the reference hospital.

US-FNAC was performed by two experienced endocrinologists. Each nodule was aspirated at least twice, using a 22- or 23-gauge needle attached to a 10 mL syringe and a Cameco^®^ ((Cameco AB, Taby, Sweden) syringe holder. Direct smears, usually 8 to 10, were stained with routine stains (air-dried smears and Giemsa stain) and 2 to 3 smears were placed in alcohol and Papanicolaou stain [29,30]. Cytology smears were interpreted in accordance with Bethesda classification 2017. The series in this study was composed of one US-FNAC and its corresponding formalin-fixed paraffin-embedded (FFPE) tissue from each nodule. Then, 3 μm-thick sections of FFPE samples for each case were used for hematoxylin-eosin (H&E) staining. This procedure is based in 5 steps: dewaxing paraffin sections, hydration, coloration with hematoxylin and eosin, dehydration, and diaphanization.

All the samples were reviewed by two independent pathologists based on the 4th edition of the World Health Organization (WHO) classification of tumors of endocrine organs [31]. Two subgroups were separately analyzed, the indeterminate nodules corresponding to Bethesda categories III and IV and the cases diagnosed as PTCs, to obtain a homogeneous group.

### 2.2. Methodology

#### 2.2.1. DNA Extraction

To extract genomic DNA, 10 µm sections of formalin-fixed paraffin-embedded (FFPE) tissues were manually micro dissected using H&E guidance, and the GRS Genomic DNA BroadRange Kit (GRiSP Research Solutions, Lisbon, Portugal) was used. The smears from US-FNAC slides were scraped to a tube, and the QIAmp^®^ DNA Investigator Kit (Qiagen, Germantown, MD, USA) was used for DNA extraction from the US-FNAC according to the manufacturer’s instructions. A NanoDropTM One UV185 Vis Spectrophotometer (Thermo Fisher Scientific Inc., Waltham, MA, USA) was used for quantification of isolated DNA.

#### 2.2.2. Mutational Screening

PCR and Sanger sequencing were used for genetic characterization of tumors regarding *TERTp*, *BRAF*, and *RAS* (*NRAS*, *HRAS*, and *KRAS*) mutations, with primer design accounting for the most frequent regions mutated on thyroid carcinoma, namely *TERTp* -124 and -146 promoter regions, *BRAF* exon 15 (codon 600), *NRAS* exon 2 (codon 61), *HRAS* and *KRAS* exons 1 and 2 (codons 12, 13, and 61) [32,33].

A Multiplex PCR kit (QIAGEN, USA) and Bioline PCR kit (MyTaq HS Mix 2X, Memphis, TN, USA) were used for DNA amplification following the manufacturers’ instructions. Sanger sequencing using the ABI Prism Big Dye Terminator kit v3.1 Cycle Sequencing and capillary electrophoresis using the Applied Biosystems (Waltham, MA, USA) 3130/3130xl Genetic Analyzers were performed. All detected mutations were validated by performing a new independent analysis.

#### 2.2.3. Clinicopathological Features

All primary lesions were evaluated, both concerning the patient’s characteristics (age and gender), as well as the tumor’s clinicopathological features: tumor size, extra thyroidal invasion, capsule invasion, vascular and lymphatic invasion, presence of fibrosis, inflammatory infiltrate, tall cell, oncocytic component, psammoma bodies, calcification, necrosis, and presence of lymph node metastases, focality, and laterality. Due to the reduced number of some histological subtypes of DTC, only PTCs were considered in the statistical analysis of the clinicopathological features of the patients.

#### 2.2.4. Statistical Analysis

The analysis was performed with categorical variables presented with frequencies (percentages) and quantitative variables with mean, standard deviation (SD), and range (minimum–maximum). Regarding quantitative variables, Student’s *t* and non-parametric Mann–Whitney tests were used to compare groups, as appropriate. In the case of categorical variables, the Chi-square or Fisher’s exact tests were used, as required.

To assess the strength of cyto-histological genetic profile agreement, Cohen’s Kappa was estimated and interpreted according to Altman (1999) [34]. The discriminative ability of genetic biomarkers regarding malignancy was performed by estimating sensitivity, specificity, and positive and negative predictive values with corresponding 95% confidence intervals. A level of significance α = 0.05 was considered. Data analysis was performed using the SPSS software version 27.0 (IBM Corp. Released 2020. IBM SPSS Statistics for Windows, Version 27.0. Armonk, NY, USA: IBM Corp).

## 3. Results

### 3.1. Series Description

#### 3.1.1. Epidemiologic Data

Of the 259 patients, 209 (80.7%) were female, with mean age of 53 (15.8) years (range 18–84), and 50 (19.3%) were male, with mean age of 54 (15.9) years (range 23–81). The distribution of age at diagnosis was similar across categories of gender (*p* = 0.890) and across categories of histological diagnosis (*p* = 0.186).

Mean tumor size was 33.5 mm (10.79) for benign lesions and 27.6 mm (15.57) for malignant lesions; the distribution of tumor size was not the same across categories of histologic diagnosis, malignant tumors being smaller than benign lesions (*p* < 0.001).

The 259 histology samples were composed of 50 (19.3%) benign lesions (control) and 209 (80.7%) malignant lesions (cases). PTCs represented the larger group in malignant histology, with 182 (87.1%) cases.

The matched 259 cytology samples were distributed according to the Bethesda classification, as follows: I. non-diagnostic (ND), 15 samples (5.8%); II. benign (B), 47 samples (18.1%); III. atypia of undetermined significance (AUS), 43 samples (16.6%); IV. follicular neoplasm (FN), 58 samples (22.4%); V. suspicious for malignancy (SM), 42 samples (16.2%); and VI. malignant, 54 samples (20.9%).

The distribution of the cytology samples within each histological subtype is presented in Table 1.

The 39% of nodules whose cytological result was indeterminate corresponded to histology with 23 benign (22.8%) and 78 malignant lesions (77.2%).

#### 3.1.2. Cyto-Histological Genetic Profile

The molecular status of *TERTp*, *BRAF*, and *RAS* (*NRAS*, *HRAS*, and *KRAS*) genes of the cytologies and histologies in all series is summarized in Table 2. Mutations were present in 85 cytologies (32.8%) and in 130 histologies (50.2%).

The mutation frequencies observed in cytology and histology specimens within our series were, respectively, *TERTp*: 3.7% vs. 7.9%; *BRAF*: 19.5% vs. 25.1%; *NRAS*: 4.4% vs. 7.9%; *HRAS*: 4.8% vs. 7.6%; and *KRAS*: 1.6% vs. 2.8%.

Mutation frequencies in cytology and histology samples based on histology subtypes are show in Table 3, being detected in PTCs in 97.6% (83/86) of the cytology and in 94.6% (122/129) of the histology specimens.

*TERTp* and *BRAF* mutation was present only in PTCs, both in cytology and in histology. *RAS* mutation was present in cytology and histology, respectively in 25 and 38 PTCs, in 1 and 3 benign cases, in 2 and 1 WDT-UMD cases, only 1 histology sample of Follicular thyroid carcinoma (FTC), and one non-invasive follicular thyroid neoplasm with papillary-like nuclear features (NIFTP). In Hürthle cell carcinoma (HCC), no mutations were identified.

Genetic mutations in cytology samples based on Bethesda categories are presented in Appendix A.

*BRAF* mutations represented most mutations, 43 cases, with 5 cases in Bethesda III and IV, 8 cases in Bethesda V, and 30 cases in Bethesda VI. *RAS* mutations were present in 26 cases, divided into the different Bethesda categories. Concomitant mutation of *TERTp*/*BRAF* (six cases) or *TERT*/*RAS* (one case) were always identified in cases with malignant histology but with variable cytology: one in B-III, two in B-IV, and four in B-VI. No mutations were found in the Bethesda ND category.

### 3.2. Genetic Alterations in Indeterminate Nodules

Of the 259 cases, 39% presented cytology of indeterminate nodules (101/259), with 43 AUS (42.6%) cases and 58 FN (57.4%). Among the indeterminate nodules, 23 (22.8%) were classified as benign lesions in histology, while 78 (77.2%) were classified as malignant in histology.

Of the 78 malignant cases with indeterminate cytology, 66 were PTCs in histology. The remaining 12 indeterminate nodules were distributed among the other types of DTC cases (Table 1).

Twenty-five (24.8%) indeterminate nodules presented mutations in cytology samples, with one *RAS* mutation in a benign lesion and twenty-four in malignant tumors, being thirteen (17.3%) for *RAS* mutation, seven (9.3%) for *BRAF*, and four (5.6%) for *TERTp* mutation.

In the histology samples of the indeterminate nodules, 48 (47.5%) were mutated, being two benign lesions with *RAS* mutation and in 46 malignant tumors, of which 11 (14.5%) were found to have *TERTp* mutation, 13 (16.9%) had *BRAF*, and 22 (29.3%) had *RAS* mutation. Concomitant *TERTp* and *BRAF* mutations were identified in three cases.

Mutation frequencies in cytology and histology samples of indeterminate nodules were, respectively, *TERTp*: 4.3% vs. 11.1%; *BRAF*: 7.2% vs. 13%; and *RAS*: 14.4% vs. 24.5% (Table 4).

The molecular status of *TERTp*, *BRAF*, and *RAS* (*NRAS*, *HRAS*, and *KRAS*) genes, including mutation types in cytologies and histologies for indeterminate nodules, is summarized in Appendix A.

### 3.3. Papillary Thyroid Carcinomas

#### 3.3.1. Clinicopathological Features of PTCs

For statistical analysis of the clinicopathological features in our series, only PTCs were considered due to the limited number of the other malignant subtypes of DTC. From the 209 patients with malignant histology of DTCs, 182 (87.1%) were PTCs, and 27 (12.9%) were other subtypes of DTC (Table 3).

Of the 182 PTC tumors, 150 (82.4%) were females; the mean age at diagnosis was 52 (15.9) years (range 18–84); mean tumor size was 26.7 mm (15.38) (*p* < 0.001).

The histological cases of PTCs encompassed different cytological diagnoses. The distribution of samples according to Bethesda categories was as follows: I. (ND) 12 (6.6%); II. (B) 16 (8.8%); III. (AUS) 30 (16.5%); IV. (FN) 36 (19.8%); V. (SM) 35 (19.2%); and VI. (M) 53 (29.1%).

The histological distribution of PTC variants and the corresponding clinicopathological features are presented in Appendix A.

#### 3.3.2. Cyto-Histological Genetic Profile in PTCs

In PTCs, mutation frequencies in cytology and histology were, respectively, *TERTp*: 5.3% (9) vs. 10.8% (19); *BRAF*: 27.4% (48) vs. 35.6% (63); and *RAS*: 14.3% (25) vs. 22.1% (38).

Concomitant *TERTp* and *BRAF* mutations were identified in 10 PTCs, in cytology and/or histology samples.

The mutations in cytology and histology samples through histology subtypes of PTCs are shown in Appendix A.

#### 3.3.3. Relationship between the Clinicopathological Features and the Genetic Profile in PTCs

The associations between the clinicopathological features and the genetic alterations of PTCs are presented in Appendix A. No statistically significant differences were achieved for gender or age. The distribution of tumor size was different across types of mutations at diagnosis: tumors with *BRAF* mutations were smaller (*p* < 0.001) than tumors without mutation.

The presence of *TERTp* mutations was significantly associated with vascular invasion (*p* = 0.004) and with an oncocytic component (*p* = 0.017). *BRAF* mutations showed a significant association with the presence of extra thyroidal invasion, capsule invasion, vascular invasion, lymphatic invasion, oncocytic component, fibrosis, inflammatory infiltrate, tall cell component, psammoma bodies, calcification, focality, and LNM (*p* < 0.001 for all). Concomitant *TERTp* and *BRAF* mutations were significantly associated with extra thyroidal invasion, capsular invasion, vascular and lymphatic invasion, tall cells, and oncocytic component (*p* < 0.001 for all).

Concerning *RAS* mutations, cases exhibiting *NRAS* mutations demonstrated a significant association with the presence of a capsule (*p* = 0.030) and unifocality (*p* = 0.030). No significant associations were observed between the presence of *HRAS* or *KRAS* mutations and any clinicopathological features.

### 3.4. Statistical Analysis of Cyto-Histological Profile

The cyto-histological agreement achieved for molecular alterations in our series is summarized in Table 5. The comparison between cytology and histology for molecular alterations was possible in 244 cases. The cyto-histological concordance was observed in 94.9% of cases, with a Cohen’s k = 0.67, which is considered substantial agreement (0.60–0.80). The concordance obtained in PTCs was 94.6% with substantial cyto-histological agreement (k = 0.659).

The agreement of the molecular testing in cytological and histological samples was evaluated in 95 indeterminate nodules, with concordance in 95.6% of cases, and a k = 0.643 was also considered as substantial agreement.

When analyzing the genes independently, the genes showing cyto-histological molecular agreement that was considered substantial were *BRAF* and *HRAS* in DTC, in PTC, and in indeterminate nodules. *NRAS* and *TERTp* genes presented a cyto-histological molecular agreement considered moderate (0.40–0.59). *KRAS* mutation was excluded from this analysis due the small number of cases.

The mutations’ discriminative ability for the diagnosis of malignancy in DTCs, PTCs, and indeterminate nodules is shown in Table 6 for all series and in Appendix A for each group.

The *TERTp* and *BRAF* mutations exhibited a specificity of 100% both in cytology and in histology, with a positive predictive value (PPV) of 100% in all the series. *RAS* gene mutations demonstrated a specificity of 98% in cytology and 94% in histology for DTCs and PTCs, with PPVs in cytology and histology, respectively, of 96.3% and 93.3% in DTCs and 96.2% and 92.7% in PTCs; in indeterminate nodules, *RAS* presented a specificity of 95.5% in cytology (PPV 92.9%) and 91.3% in histology (PPV 91.7%). However, the sensitivities and the negative predictive value (NPV) of all mutations were very low in our series for DTCs, PTCs, and indeterminate nodules.

### 3.5. Cyto-Histologic Molecular Discordant Cases

Cyto-histologic molecular discordant cases were evaluated, as shown in Appendix A.

The discordance for the presence of mutations in the BRAF gene was present in 21/255 cases, representing 8.2%; for the TERTp gene it was present in 15/254 cases, with 5.9% discordance; for NRAS, HRAS, and KRAS genes, the discordance was 5.1%, 3.6%, and 3.6%, respectively.

The genetic cyto-histologic discrepancy was analyzed more closely in these cases. In the presence of mutations in the BRAF gene, discordance was present in 21 cases, being 3 cases in cytology (1 B-VI, 1 B-V, and 1 B-I) that were negative in histology and 18 cases in histology (8 c-PTC, 5 FC-PTC, and 3 others in PTC) that were negative in cytology. In the TERTp gene, discordance was shown in 15 cases, with 2 cases being positive in cytology (B-VI) that were negative in histology and 13 in histology (6 FC-PTC, 5 c-PTC, and 3 others in PTC). For NRAS genes, discordance was present in 13 cases, 2 in cytology (1 B-VI and 1 B-IV) and 11 in histology, 1 benign and 10 malignant (9 PTC and 1 FTC); for HRAS genes, discordance occurred in 9 cases, 1 in cytology (B-VI) and 8 with malignant histology (6 FC-PTC, and 2 others in PTC). And, for KRAS genes, there were nine cases, three in cytology (B-V, B-III, and B-II) and six in histology (one benign, one NIFT, three FC-PTC, and one c-PTC).

## 4. Discussion

Molecular tests have been proposed to be used in clinical practice for diagnosis (pre-surgical markers) and follow-up of thyroid nodules. The 2017 ETA guidelines review methodological aspects and limitations of molecular diagnoses using US-FNAC for thyroid nodules [18]. In recent years, panels of somatic genetic alterations have been proposed as a potential approach. However, due to the current limitations in accurately predicting the malignancy of thyroid nodules, most authors consider that further studies with larger sample numbers, rigorous normalization techniques, and results validation are required before their widespread adoption in clinical practice [35,36].

The present study was designed to contribute to clarification of some of these challenges. A series of 259 surgical resected thyroid nodules, 209 with a histologic diagnosis of malignancy and 50 with benign diagnosis (as the control), were selected, and the matched pre-surgical cytologic diagnostic slides were collected. Then, we evaluated the genetic alterations in the 518 matched histological/cytological samples. The strength of our study is the relatively large number of matched cytology/histology samples included. This less-usual study design also implies that our series would have a significant number of malignant tumors in the Bethesda categories III and IV (please see limitations of the study below), resulting from the series selection criteria. But that design is also advantageous, allowing more accurate evaluation regarding to what extent genetic markers can improve cytology pre-surgical results.

*BRAF* mutations were predominant in our series, either in cytology or histology samples, followed by *RAS* mutations, reflecting the composition of the series that was enriched in PTC and in low-risk tumors. In accordance with other published studies [37,38], including a study from Whitney S. Goldner [39], the majority of *BRAF* mutations were p.V600E. *TERTp* mutations were assessed in two different hotspots, and most mutations in cytologies and histologies were found in the −124 position, as previously reported regarding thyroid tumors [40]. Tumors in the malignant category (Bethesda VI) presented a high frequency of *BRAF* mutations, followed by concomitant *TERT* and *BRAF*, as well as *TERT* and *RAS* mutations.

Our first approach was to evaluate cyto-histological genetic profile agreement for molecular alterations with special attention paid to the nodules that have been diagnosed as indeterminate cytological diagnoses, a category that raises important diagnostic doubts.

An acceptable cyto-histological genetic profile agreement for molecular alterations was attained in the overall series, indicating a good level of consistency between the two types of samples in the genetic analysis. *BRAF* and *HRAS* were the genes showing substantial cyto-histological molecular agreement, whereas moderate agreement was found for *NRAS* and *TERTp* genes. In each case, the divergent molecular results between cytology and histology can reflect tumor heterogeneity, multifocality, and/or low allelic frequencies, the repercussion of these factors being cytology samples that present a lower percentage of (neoplastic) cells that could increase the number of false negative results.

Then, we evaluated if the mutational status of cytology, particularly for lesions categorized as indeterminate (AUS and FN), can enhance diagnostic ability and improve patient management [41]. In cytology, the frequency of the mutations increased from a lower Bethesda category (B and AUS) to a higher Bethesda category (SM and M). Of note, in the control cases (50 selected histological benign lesions), only one case in cytology, and three in histology, present mutations that are all in the *RAS* genes. If we consider *TERTp*, *BRAF*, and *RAS* mutations as being in indeterminate categories (*n* = 101 cases), the cytologic results taken together with clinical, imageology, and molecular information could be improved in 25% of the cases (24 cases with mutations in any gene). Being more cautious, since *RAS* mutations are relatively frequent and were found in a few benign nodules of our series, and considering only their predictive value for malignancy of *TERTp* and *BRAF* mutations (both with a 100% specificity, 100% PPV), then 10% of the indeterminate nodules of our series (*n* = 10) could be diagnosed as malignant in a pre-surgical phase, without performing more FNACs and allowing better surgical options and health cost reduction.

Indeterminate nodules represented a heterogeneous group. The high number of malignant cases in our series result, as mentioned above, from two causes: on one hand because case selection was conducted based on a histologic diagnosis of malignancy (or benignity in the 50 control cases), and on the other hand these patients underwent surgery due to clinical reasons besides FNAC results (e.g., US features). A good cyto-histological genetic profile agreement for molecular alterations was achieved between the two types of samples in indeterminate nodules, reinforcing the role of molecular analyses before surgery in those cases. *RAS* were found to be the most frequently mutated genes in the indeterminate nodules, in accordance with Censi S. [42], who evaluated a large cohort of indeterminate thyroid nodules and detected *RAS* mutations in 18% of all series.

Our series was particularly enriched in PTCs, which allowed us to evaluate the clinicopathological implications of the genetic background of the tumors, as evaluated via the molecular analysis of the histologies. Regarding the molecular results of the PTCs, we found a prevalence of mutations in BRAF and TERT genes like those reported by Liu R. [43] and Insilla A. [44], but it was lower than that in some studies [45].

Several statistically significant associations between the clinicopathological and molecular features of the PTCs were found in our series. BRAF mutations were significantly associated with several clinicopathological features, in accordance with Liu, X. [46]. Despite the unfavorable presence of BRAF p.V600E in PTCs, its prognostic role remains debated, as mentioned by Scheffel et al. [47]. TERTp mutations were significantly associated with features of the worst prognoses, as mentioned by Bournaud et al. [48]. On the contrary, RAS mutations were associated with the presence of capsules and a better outcome, as reported in previous study [49].

Xing, M. et al. [50] reported that 6.9% of all PTCs have concomitant mutations in *TERTp* and *BRAF* genes, which were significantly associated with clinicopathological features of worst prognoses and tumor progression, namely LNM, extra thyroidal, and vascular invasion. Estrada Flórez A. P. et al. [51] published similar results, with concomitant *TERTp* and *BRAF* mutations in 10% of their series and the same significant associations. In accordance with the abovementioned studies, we observed concomitant *TERTp* and *BRAF* mutations in 5.6% of PTCs, significantly associated with clinicopathological features related to tumor aggressiveness. On the other hand, Melo M. et al. [52] have not found this association in their series. Ren H. et al. [9] have found no associations between the presence of both mutations and LNM, but they found a significant association with extra thyroidal invasion, large tumors, and older patients. No concomitant mutations were observed for *BRAF* and *RAS* as expected since these mutations are described as mutually exclusive events [53].

The *BRAF* and/or *TERTp* mutations were only present in malignant cases, in both cytology and histology samples, which we could consider as “rule in” tests, in accordance with American Thyroid Association (ATA) guidelines (PPV > 95%) [54,55]. *RAS* mutations presented slightly lower specificity in our series due to their presence in some benign cases and low-grade tumors. However, *RAS* mutation showed a higher specificity when compared with previous reports, resulting from the selection of cases in our series. The sensitivities and NPVs of all mutations were much lower in both cytology and histology samples, not excluding malignancy within our series, as reported in other studies [56,57].

The low-risk neoplasms NIFTP and WDT-UMP tumors represented a few cases in our series and presented only *RAS* mutations, in accordance with others [58].

Our study presented low percentages of cyto-histologic molecular discrepancies and good global agreement. The cyto-histologic molecular discrepancies resulted mainly from two causes: (i) the impossibility of repeating US-FNAC when cytology samples presented a lower percentage of cells and (ii) the selection of representative histology tissues that do not correspond to the same nodule that was subjected to the punction.

This study, conducted in a European population, presented limitations namely due to its retrospective nature, which is the only way to obtain a large series of operated thyroid nodules and to perform a matching cyto-histological comparative analysis of the genetic profiles in DTCs. The small number of genetic mutations analyzed, and the detection limit of Sanger sequencing used in this study, could be considered potential limitations, especially when compared to the extensive mutation panels available [59,60] as well as other sequencing methods. Due to the retrospective nature of our study, it was not possible to obtain RNA from cytology slides (some of them stored for long periods) and to obtain information about the prevalence of fusion genes (namely RET/PTC) in the series, as reported in previous studies [15,61]. However, our aim was to assess the reliability of molecular diagnoses obtained via US-FNAC in thyroid nodules using a laboratory method that offers practical replicability, is widely available, and can be conducted at low cost. By focusing on a smaller set of mutations, we assessed the confidence level of our findings and provided valuable insights within the constraints of our study design.

Our results obtained via cyto-histological comparative analysis of the genetic profiles in thyroid nodules reinforce the potential clinical utility of molecular testing in cytological assessments using US-FNAC, contributing to the anticipation of the genetic profiles of the tumors and their biological behavior. By incorporating genetic information, clinicians can make better-informed decisions; optimize treatment strategies, including active surveillance and surgical options; and provide better care for patients with ambiguous cytological findings.

## 5. Conclusions

Our study yielded compelling results, demonstrating a good agreement between cytological and histological findings for molecular alterations in both DTCs and in indeterminate nodules. This minimally invasive and cost-effective approach has the potential to enhance diagnostic accuracy and streamline patient management by integrating molecular testing into routine clinical practice. While the genetic profile’s ability to exclude malignancy is limited, it effectively confirms malignancy in cytology samples, reducing repeated US-FNACs and contributing to referrals of patients towarddiagnostic, including surgical options. Moreover, the numerous associations identified between clinicopathological features and genetic profiles of PTCs suggest that molecular analysis using US-FNAC can provide early insights into tumor genetic profile and behavior. To gain a comprehensive understanding of tumor evolution and assess the applicability of molecular tests in the European population, further studies are needed that will shed new light on the dynamics of these tumors and their genetic characteristics.

## Figures and Tables

**Table 1 diagnostics-14-00278-t001:** Distribution of the cytology samples within histological subtypes in all series.

Cytology Diagnosis *n* = 259	Histology Diagnosis *n* = 259						
	Benign	WDT-UMP	NIFT	PTC	FTC	HCC	Total
1. ND	00 (0%)	2 (0.8%)	2 (0.8%)	11 (4.2%)	0 (0%)	0 (0%)	15 (5.8%)
2. Benign	25 (9.7%)	1 (0.4%)	3 (1.2%)	15 (5.8%)	2 (0.8%)	1(0.4%)	47 (18.1%)
3. AUS	12 (4.6%)	1 (0.4%)	0 (0%)	30 (11.6%)	0 (0%)	0 (0%)	43 (16.6%)
4. FN	11 (4.2%)	5 (1.9%)	1 (0.4%)	36 (13.9%)	2 (0.8%)	3 (1.2%)	58 (22.4%)
5. SM	02 (0.8%)	1 (0.4%)	0 (0%)	35 (13.5%)	3 (1.2%)	1 (0.4%)	42 (16.2%)
6. Malignant	00 (0%)	1 (0.4%)	0 (0%)	53 (20.5%)	0 (0%)	0 (0%)	54 (20.9%)
Total	50 (19.3%)	11 (4.2%)	6 (2.3%)	180(69.5%)	7 (2.7%)	5 (1.9%)	259 (100%)

Legend: ND: non-diagnostic, B: benign, AUS: atypia of undetermined significance, FN: follicular neoplasm. SM: suspicious for malignancy; and M: malignant. WDT-UMP: well-differentiated thyroid tumor of uncertain malignant potential; NIFT: non-invasive follicular thyroid neoplasm with papillary-like nuclear features; PTC: papillary thyroid carcinoma; FTC: follicular thyroid carcinoma; HCC: Hürthle cell carcinoma.

**Table 2 diagnostics-14-00278-t002:** Genetic mutations in cytology and histology in all series.

Genetic Mutations	Cytology			Histology		
	*n* *	Mutated *n* (%)	Mutation Type	*n* *	Mutated *n* (%)	Mutation Type
*TERTp*	246	09 (3.7)	08 (−124 G>A) 01 (−146 G>A)	254	20 (7.9)	13 (−124G>A) 07 (−146 G>A)
*BRAF*	251	49 (19.5)	48 (p.V600E) 01 (p.K601E)	255	64 (25.1)	62 (p.V600E) 02 (p.K601E)
*NRAS*	250	11 (4.4)	11 (p.Q61R)	254	20 (7.9)	20 (p.Q61R)
*HRAS*	250	12 (4.8)	07 (p.Q61R) 05 (p.Q61K)	251	19 (7.6)	12 (p.Q61R) 06 (p.Q61K) 01 (p.G13A)
*KRAS*	250	04 (1.6)	04 (p.Q61R)	251	07 (2.8)	05 (p.Q61R) 01 (p.G12A) 01 (p.G12R)
Total		85 (32.8)			130 (50.2)	

* Molecular results were not conclusive in all samples due to technical issues. Legend: *TERTp*: telomerase reverse transcriptase promoter; *BRAF*: B-Raf proto-oncogene, serine/threonine kinase; *NRAS*: NRAS proto-oncogene, GTPase; *HRAS*: HRas proto-oncogene, GTPase, and *KRAS*—*KRAS* proto-oncogene, GTPase.

**Table 3 diagnostics-14-00278-t003:** Mutation frequencies in cytology and histology samples by histology subtypes.

Mutations	Histology Subtypes *n* = 259						
	Benign *n* = 50	WT-UMD *n* = 10	NIFT *n* = 6	PTC *n* = 176	FTC *n* = 7	HCC *n* = 5	Total
*TERTp*							
H *n* = 254	0	0	0	20	0	0	20 (7.9%)
C *n* = 246	0	0	0	9	0	0	9 (3.7%)
*BRAF*							
H *n* = 255	0	0	0	64	0	0	64 (25.1%)
C *n* = 251	0	0	0	49	0	0	49 (19.5%)
*RAS*							
H *n* = 251	3	2	1	38	1	0	45 (17.9%)
C *n* = 250	1	1	0	25	0	0	27 (10.8%)
Total							
Histology	3	2	1	122	1	0	129 (100%)
Cytology	1	1	0	83	0	0	85 (100%)

Legend: H: histology; C: cytology; WDT-UMP: well-differentiated thyroid tumor of uncertain malignant potential; NIFT: non-invasive follicular thyroid neoplasm with papillary-like nuclear features; PTC: papillary thyroid carcinoma; *TERTp*: telomerase reverse transcriptase promoter; *BRAF*: B-Raf proto-oncogene, serine/threonine kinase; *RAS*: RAS proto-oncogene, GTPase.

**Table 4 diagnostics-14-00278-t004:** Genetic mutation in cytology and histology of indeterminate nodules.

Genetic Mutations		Cytology (Mutated) *n* = 25 (24.8%)			Histology (Mutated) *n* = 48 (47.5%)			
Histology	*n*	Benign *n* = 23	Malignant *n* = 78	Total *n* = 101	*n*	Benign *n* = 23	Malignant *n* = 78	Total *n* = 101
*TERTp*	94	0 (0%)	4 (5.6%)	4 (4.3%)	99	0 (0%)	11 (14.5%)	11 (11.1%)
*BRAF*	97	0 (0%)	7 (9.3%)	7 (7.2%)	100	0 (0%)	13 (16.9%)	13 (13%)
*RAS*	97	1 (1%)	13 (17.3%)	14 (14.4%)	98	2 (8.7%)	22 (29.3%)	24 (24.5%)

Legend: *TERTp*: telomerase reverse transcriptase promoter; *BRAF*: B-Raf proto-oncogene, serine/threonine kinase; *RAS*: RAS proto-oncogene, GTPase.

**Table 5 diagnostics-14-00278-t005:** The cyto-histological agreement in DTCs, PTCs, and indeterminate nodules.

	All Series *n* = 259		PTCs *n* = 182		Indeterminate Nodules *n* = 101	
	Concordance (%)	Cohen’s k	Concordance (%)	Cohen’s k	Concordance (%)	Cohen’s k
Genes (Total)	94.9%	0.670	94.6%	0.659	95.6%	0.643
*TERTp*	94.6%	0.493	94.5%	0.512	94.6%	0.591
*BRAF*	92.7%	0.790	91.9%	0.781	94.8%	0.710
*NRAS*	95%	0.576	95.5%	0.620	95.8%	0.579
*HRAS*	97%	0.744	96.8%	0.724	93.7%	0.695

Legend: DTCs: Differentiated Thyroid Carcinomas, PTCs: Papillary Thyroid Carcinomas. *TERTp*: telomerase reverse transcriptase promoter; *BRAF*: B-Raf proto-oncogene, serine/threonine kinase; *NRAS*: NRAS proto-oncogene, GTPase; *HRAS*: HRas proto-oncogene, GTPase; Cohen’s k: Cohen’s Kappa value: to interpret the strength of the agreement (Altman, 1999).

**Table 6 diagnostics-14-00278-t006:** The discriminative ability of mutations in DTCs, PTCs, and indeterminate nodules.

Mutations	DTCs *n* = 209			PTCs *n* = 180			Indeterminate Nodules *n* = 101		
	Se (%)	Sp (%)	PPV (%)	Se (%)	Sp (%)	PPV (%)	Se (%)	Sp (%)	PPV (%)
*TERTp*									
Histology	9.8	100	100	10.8	100	100	14.5	100	100
Cytology	4.6	100	100	5.26	100	100	5.56	100	100
*BRAF*									
Histology	31.2	100	100	35.6	100	100	16.9	100	100
Cytology	24.3	100	100	27.4	100	100	9.33	100	100
*RAS*									
Histology	21	94	93.3	22.1	94	92.7	29.3	91.3	91.7
Cytology	12.9	98	96.3	14.3	98	96.2	17.3	95.5	92.9

Legend: DTCs: differentiated thyroid carcinomas; PTCs: papillary thyroid carcinomas; Se: sensitivity; Sp: specificity; PPV: positive predictive value; *TERTp*: telomerase reverse transcriptase promoter; *BRAF*: B-Raf proto-oncogene, serine/threonine kinase; *RAS*: RAS proto-oncogene, GTPase.

## Data Availability

Data are contained within the article and Appendix A.

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
