# Peer review of "Comparative Cyto-Histological Genetic Profile in a Series of Differentiated Thyroid Carcinomas"

_diagnostics, 2024, doi:10.3390/diagnostics14030278_

Round 1
Reviewer 1 Report
Comments and Suggestions for Authors
The authors did not used the results of similar study for disscussion for BRAF mutation incidence see Pekova et al and/Bulanova et al. The used method are necessery for surgery or treatment protocol see ATA and ETA rules abd descalation protocols as well. The significance of anz othe fusion genes is low no correlation Q. Were used.
Author Response
First Reviewer:
1.The authors did not used the results of similar study for discussion for BRAF mutation incidence see Pekova et al and/Bulanova et al.
Answer 1:
Thank you for your remark. We have modified the introduction and add additional references, as suggested, with additional studies reporting molecular characterization of thyroid tumors.
Please refer to introduction:
“Despite DTC favorable prognosis, it was advanced that clinicopathological features and the presence of some genetic alterations can influence their progression [9]. Pre-surgical accurate diagnosis is crucial to decide the optimal medical or surgical treatment [10].”
…..
“Genetic changes in genes encoding cell signaling pathways determine tumors development, including in DTCs. The main known genetic causes of thyroid cancer include point mutations in the BRAF, RAS, TERT, RET and TP53 genes and the fusions genes RET/PTC, PAX8/PPAR-Y, and NTRK. Numerous studies have been conducted to identify genetic or molecular biomarkers that facilitate the pre-surgical diagnosis (in categories ND, III and IV) and prognosis of thyroid cancer, but current results remain inconclusive and without definitive evidence [15,16].”
Please refer to Discussion:
“Several statistically significant associations between the clinicopathological and mo-lecular features of the PTCs were found in our series. BRAF mutations were signifi-cantly associated with several clinicopathological features, in accordance with Liu, X. (2018) [45]. Despite the unfavorable presence of BRAF p.V600E in PTCs, its prognostic role remains debated, as mentioned by Scheffel et al [46].”
New referencies added – [10] and [15] and [46]
2.The used method are necessery for surgery or treatment protocol see ATA and ETA rules abd descalation protocols as well.
Answer 2:
Thank you for your note. In fact, we followed ETA and ATA guidelines, but that information was not clear in the Ms. We clarify that in a modified paragraph in Materials and Methods:
“This retrospective study was conducted in a series of patients with thyroid nodular disease suspicion of malignancy that underwent surgery, in a single non oncologic hospital between 2013 and 2020. Criteria of suspicion and/or malignancy in thyroid nodules were based in clinic, ultrasound features and Bethesda classification for cytology (risk of malignancy) [11], in accordance with 2017 ETA and 2015 ATA guidelines. Recommendations on the extent of surgical treatment and surgical protocols used also followed these guidelines [18,19].”
We also, extend the description of the methodology concerning US-FNAC.
“US-FNAC was performed by two experienced endocrinologists, using a standardized technique [29,30]. Each nodule was aspirated at least twice, using a 22 or 23-gauge needle attached to a 10 mL syringe, and a Cameco ® syringe holder. Direct smears, usually 8 to 10 were stained with routine stains (air dried smears and Giemsa stain) and 2 to 3 smears were placed in alcohol and Papanicolau stain). Cytology smears were interpreted in accordance with Bethesda classification 2017. The series in this study was composed by one US-FNAC and corresponding formalin-fixed paraffin-embedded (FFPE) tissue from each nodule.”
And we add new references as suggested by Reviewer R1 – [29,30]
3.The significance of anz othe fusion genes is low no correlation Q. Were used.
Answer 3:
Thank you for your comment. We were not sure if we get your full comment, since it looks truncated, and we ask to the Journal staff if there was something missing, but we were informed that it was complete. So, we are not sure if we get the Reviewer point.
Our study was retrospective, and we used pre-stained cytology slides, some of them stored for a long period, and FFPE material. This limited the obtention of RNA in paired cyto-histologic samples and, consequently, the possibility to study fusion genes. However, we add a sentence in the Ms. referring this limitation of our study.
“Due to the retrospective nature of our study, it was not possible to obtain RNA from cytology slides (some of them stored for long periods) and obtain information about the prevalence of fusion genes (namely RET/PTC) in the series, as reported in previous studies [15,56].”
And we add new reference as suggested by Reviewer R1 – [56].
Reviewer 2 Report
Comments and Suggestions for Authors
1) In this study, the authors aimed to compare the cyto-histological genetic profile, and to evaluate the reliability of molecular tests on US-FNAC in accurately diagnosing DTCs and predicting biologic behavior of papillary thyroid carcinomas (PTCs). The results suggested that molecular analysis in US-FNAC may anticipate the genetic profile and the behavior of thyroid tumors, confirming malignancy and contributing to refer patients towards surgery. It was interesting that the overall cyto-histological agreement of the genetic mutations was high, especially in indeterminate nodules (95.7%).
2) Line 196-198: exactly as the author showed, the mutation frequencies of TERTp, BRAF, NRAS, HRAS and KRAS were all lower in cytology when compared with that in histology. Though, the overall cyto-histological agreement of the genetic mutations was higher than cytology/histology alone, please supply the data concerning the inconsistent cyto-histological cases.
3) To develop further results, I suggest that a subgroup analysis is needed, which mainly focus on different DTC subtypes.
4) Please detailed describe the aspiration procedure, as a single puncture could absolutely acquire less cell components in compared with multiple punctures. Therefore, the nuclei acid amplification results could have changed due to the tumor heterogeneity. I’m curious of the discrepancies under this condition.
Comments on the Quality of English LanguageNon.
Author Response
Second Reviewer:
Thank you very much for the time and attention spend in the revision of our Ms. We consider your suggestions that have helped to improve our Ms.
1) In this study, the authors aimed to compare the cyto-histological genetic profile, and to evaluate the reliability of molecular tests on US-FNAC in accurately diagnosing DTCs and predicting biologic behavior of papillary thyroid carcinomas (PTCs). The results suggested that molecular analysis in US-FNAC may anticipate the genetic profile and the behavior of thyroid tumors, confirming malignancy and contributing to refer patients towards surgery. It was interesting that the overall cyto-histological agreement of the genetic mutations was high, especially in indeterminate nodules (95.7%).
Our comment:
Thank you very much for your comment. In fact, the aim of our study was to compare the cyto-histological genetic profile in paired cytology /histology samples of thyroid nodules, to evaluate if the molecular study could improve the cytologic diagnosis in a pre-surgical time. One of the most interesting results of our study was the demonstration that, in the undetermined group, in a real-life scenario, 10 patients could have a definitive diagnosis of malignancy at the first approach, if molecular analysis was incorporate in the diagnosis pathway. Our studies contribute to confirm that molecular analysis in US-FNAC may anticipate the genetic profile and the behavior of thyroid tumors, with an overall high cyto-histological genetic agreement.
2) Line 196-198: exactly as the author showed, the mutation frequencies of TERTp, BRAF, NRAS, HRAS and KRAS were all lower in cytology when compared with that in histology. Though, the overall cyto-histological agreement of the genetic mutations was higher than cytology/histology alone, please supply the data concerning the inconsistent cyto-histological cases.
Our answer:
As referred in the Ms. “The divergent molecular results between cytology and histology can reflect tumor heterogeneity, multifocality and/or low allelic frequencies, being the repercussion of these factors more marked on cytology samples that present a lower percentage of (neoplastic) cells that could increase the number of false negative results.”
The inconsistent cyto-histological cases can also result from the lower cellular representativity of the cytology samples that presented a lower percentage of cells (lower sensibility) and selection of the cytology slides. As referred, our study was retrospective and, in some cases, the selection of the more representative cytology slide could not correspond to the nodule analyzed by histology. Unfortunately, the retrospective match is not possible.
Nevertheless, the overall cyto-histological concordance of the genetic mutations was higher than cytology/ histology alone, as demonstrated by a lower Cohen´k in DTCs, PTCs and indeterminate nodules. However, when we obtained a cyto-histologic molecular match, the agreement is higher between cytology and histology suggesting that molecular analysis in US-FNAC may anticipate the genetic profile and the behavior of thyroid tumors.
To clarify this point, and acknowledging Reviewer suggestion, we add a new supplementary table, reporting, case-by-case, the cyto-histological molecular discordant cases.
Please refer to the Supplementary Table 7. Profile of the molecular cyto-histological discordant cases.
3) To develop further results, I suggest that a subgroup analysis is needed, which mainly focus on different DTC subtypes.
Thanks for your suggestion. We agree with you on that point but focus on different DTC subtypes was not possible with our series, due the reduced number of several DTC subtypes. Our study was retrospective and based in a general hospital consecutive series, where PTCs are the most frequent DTC, over 90% with few FTC and others. To obtain many DTCs subtypes, allowing an adequate statistical analysis, only with a much larger series, perhaps with a multicentric study. For that reason, we must opt doing subgroup analysis only in undetermined cytology and in histologic PTC.
4) Please detailed describe the aspiration procedure, as a single puncture could absolutely acquire less cell components in compared with multiple punctures. Therefore, the nuclei acid amplification results could have changed due to the tumor heterogeneity. I’m curious of the discrepancies under this condition.
Thank you for raising this interesting question. In fact, an evaluation of inconsistencies in single puncture versus multiple punctures, would be very interesting. In our study the US-FNAC was done following a standardized protocol established at our clinical center, and at least 2 aspirates are done in each nodule. Nevertheless, giving the retrospective nature of the study, as referred before, a single slide is selected from each case, and no recall of the multiple aspirates was possible. This is referred as one limitation of our study. To clarify that question, and following Reviewer request, we have added to Material and Methods section, a more detailed description of the US-FNAC technique.
“US-FNAC was performed by two experienced endocrinologists, using a standardized technique [29,30]. Each nodule was aspirated at least twice, using a 22 or 23-gauge needle attached to a 10 mL syringe, and a Cameco ® syringe holder. Direct smears, usually 8 to 10 were stained with routine stains (air dried smears and Giemsa stain) and 2 to 3 smears were placed in alcohol and Papanicolau stained. Cytology smears were interpreted in accordance with Bethesda classification 2017. The series in this study was composed by one US-FNAC and corresponding formalin-fixed paraffin-embedded (FFPE) tissue from each nodule. Three μm thick sections of FFPE samples for each case were used for Hematoxylin-eosin (H&E) staining.”
As referred before, an acceptable cyto-histological profile agreement for molecular alterations was attained in the overall series, indicating a good level of consistency between the two types of samples in the genetic analysis. The divergent molecular results between cytology and histology can reflect tumor heterogeneity, multifocality and/or low allelic frequencies, being the repercussion of these factors more marked on cytology samples that present a lower percentage of (neoplastic) cells that could increase the number of false negative results.

Round 2
Reviewer 1 Report
Comments and Suggestions for Authors
The authors improved the paper a lot.
i would like ask for any points only - methodology see the table 1,2 legend the BRAF was described like v raf protoncogen the better will be describe it by function. The Ras family include most of like Kras Hras and other subtype
protoncogenes . The quetion is what authors can present type of detected mutation KRAs 10g12 or
any historc name of protooncogenes?
the second note is if tou presented the incidence of mutation on cytology and histology can you presented the incidence of mutation in Bethesda III next column how the PTC with from this group PTC without next column benigne with mutation of the bethesda III from this column
if you done the analyse regarding to ccancer hhistology how you include nfitp and non malihgnat diagnose to cancer family with retroaktive genes analyse. The interpertetion comprae to 50 benigne is not valid if this benigne were not all of sample. This fact you can comment or this must be comment in disscusionand conslusions that your result must be present clearly and we can trust to your analyse better.
Author Response
Second round first Reviewer
1.methodology see the table 1,2 legend the BRAF was described like v raf protoncogen the better will be describe it by function. The Ras family include most of like Kras Hras and other subtype protoncogenes . The quetion is what authors can present type of detected mutation KRAs 10g12 or any historc name of protooncogenes?
Thank you for your pertinent comment. In fact, we were using the historic name of the genes. After your remark, we went to HUGO Gene Nomenclature Committee (HGNC) (https://www.genenames.org/) and we substituted for the approved name, when appropriated. In relation to RAS genes, in the instances and tables were we present the results for each individual gene (H, N, K-RAS) we corrected for the HUGO indicate name. In the Tables (or text) were we do the combined evaluation of the three RAS genes, we will refer to RAS- RAS proto-oncogene, GTPase, that, although not an approved nomenclature, it seems to us adequate in that circumstance. We hope that the Reviewer agree with the coherence of the decision.
2.the second note is if tou presented the incidence of mutation on cytology and histology can you presented the incidence of mutation in Bethesda III next column how the PTC with from this group PTC without next column benigne with mutation of the bethesda III from this column
Thank you very much for your question that allowed us to correct Table 3 and Supplementary Table 1 of our MS.
We presented mutation frequencies in cytology and histology samples by histology subtypes in Table 3, and mutational frequency through Bethesda categories in Supplementary Table 1. In the revision we noted that there was a mistake in both tables because WDT-UMP and benign tumors only presented RAS mutation, in accordance with others (please see answer to question 3).
We showed the distribution of the cytology samples within histological subtypes in all series in Table 1, where PTCs presented the highest frequencies, and the mutational status of cytology and histology samples within PTC variants in Supplementary Table 4.
We correct Table 3 and Supplementary Table 1.
3.if you done the analyse regarding to ccancer hhistology how you include nfitp and non malihgnat diagnose to cancer family with retroaktive genes analyse. The interpertetion comprae to 50 benigne is not valid if this benigne were not all of sample. This fact you can comment or this must be comment in disscusionand conslusions that your result must be present clearly and we can trust to your analyse better.
Thank you for or comment, perhaps we did not explain clearly in the MS the composition of our series. In fact, we selected, as control, 50 benign histology cases from patients with thyroid nodules submitted to surgery, regardless the cytological diagnosis. Those 50 benign histology cases were the control group when we compared cyto-histological genetic profile in all series. This means that not all have a Bethesda II diagnosis, as you can see in Table 1, but all have a confirmed histologic result of benignity.
The NIFTP was a new name for a very low risk thyroid tumor previously known as an (Encapsulated Non-invasive) Follicular Variant Papillary Thyroid Carcinoma, and was a defined entity recognized by the 2017 4th World Health Organization Classification of Tumors of Endocrine Organs29. Only in the 2022 5th WHO classification of tumors of endocrine organs, Follicular-derived thyroid tumors were categorized into “Benign tumors”, “Low-risk neoplasms” and “Malignant neoplasms” 54. Both NIFT and WDT-UMP were considered low-risk neoplasms in this 5th WHO classification.
Our series is a retrospective series, as referred, selected from 2013 to 2020. All the series was revised by an endocrine pathologist following the 2017 4th WHO Classification of Tumors of Endocrine Organs. We selected 209 malignant histology cases with differentiated thyroid carcinomas, including some that were originally classified as Follicular Variant of Papillary Thyroid Carcinoma. In the histological revision, some of them were found to be NIFTP (Non-invasive follicular thyroid neoplasm with papillary- like nuclear features) in six cases and WDT-UMP (Well differentiated thyroid tumor of uncertain malignant potential) in 11 cases . This is the reason why some cases of NIFTP and WDT-UMP tumors were analyzed . Our aim was to compare the cyto-histological genetic profile in matched cytology and histology, and we found only RAS mutations in those tumors in accordance with others 55.
Please see the Discussion where we included a sentence related with this low-risk tumors:
“The low-risk neoplasms NIFTP and WDT-UMP tumors represented a few cases in our series, and only present RAS mutations, in accordance with others [54, 55].
And two news references were added – ( 54. and 55.)
29. Lloyd RV OR, Kloppel G, Rosai J. WHO Classification of Tumors of Endocrine Organs. 4 Lyon, France: International Agency for Research on Cancer; 2017.
54. Fulvio Basolo 1 Elisabetta Macerola1 Anello Marcello Poma1 Liborio Torregrossa1 REVIEW The 5th edition of WHO classification of tumors of endocrine organs: changes in the diagnosis of follicular-derived thyroid carcinoma Endocrine (2023) 80:470–476 https://doi.org/10.1007/s12020-023-03336-4.
55.Ying-Hsia Chu and Peter M. Sadow* Noninvasive follicular thyroid neoplasm with papillary-like nuclear features (NIFTP): Diagnostic updates and molecular advances Semin Diagn Pathol. 2020 Sep; 37(5): 213–218. Published online 2020 Jun 10. doi: 10.1053/j.semdp.2020.06.001
